# VISION REMEMBER: RECOVERING VISUAL INFORMATION IN EFFICIENT LVLM WITH VISION FEATURE RESAMPLING

## ABSTRACT

The computational expense of redundant vision tokens in Large Vision-Language Models (LVLMs) has led many existing methods to compress them via a vision projector. However, this compression may lose visual information that is crucial for tasks relying on fine-grained spatial relationships, such as OCR and Chart & Table Understanding. In this paper, we propose to resample original vision features across the LLM decoder layers to recover visual information and attain efficiency. Following this principle, we introduce Vision Remember, which includes two key modules: (1) Token-Feature Cross-Attention Layer and (2) Token Bidirectional Self-Attention Layer. In the Token bidirectional attention, we employ self-attention mechanism to maintain the bidirectional interaction between vision tokens and the text-guided token. In the Token-Feature interaction attention, we introduce local cross-attention to resample the visual feature and utilize the multi-level fusion to enrich the visual representation. We conduct comprehensive experiments on multiple visual understanding benchmarks and the results with the LLaVA-NeXT baseline show that Vision Remember outperforms Token-Packer by 2.7 and FastV by 5.7 across nearly all the settings. The experimental results also validate the generalization capability of the proposed method when combined with various efficient vision projectors and LVLMs.

## 1 INTRODUCTION

In recent years, with the rapid advancement of Large Language Models (LLMs) (Achiam et al., 2023; Touvron et al., 2023; Bai et al., 2023a; Cai et al., 2024; Liu et al., 2024a), a growing body of research has focused on integrating visual parsing, understanding and generation capabilities into LLM, leading to the development of a series of Large Vision-Language Models (LVLMs) (Alayrac et al., 2022; Bai et al., 2023b; Li et al., 2023; Liu et al., 2024b;d; Lu et al., 2024). The general approach involves aligning vision tokens with the linguistic domain via a projector and then concatenating with text tokens before feeding them into an LLM.

However, vision encoders often produce a large number of vision tokens (e.g., 576 in LLaVA-1.5 (Liu et al., 2024b), max 2880 in LLaVA-NeXT (Liu et al., 2024c), and max 5760 in LLaVA-OneVision (Li et al., 2024a) for an image), which occupy the majority of the input embeddings. Due to the quadratic complexity of the attention mechanism with respect to the number of tokens, longer input embeddings consume significant computational resources and memory, impeding the applications of LVLMs in practice, particularly under computationally constrained scenarios such as edge computing and robotics.

Many existing studies try to improve the efficiency and have found that vision tokens exhibit significant redundancy (Chen et al., 2024b; Zhang et al., 2024b; Xing et al., 2024). As a result, they have made efforts to reduce the number of vision tokens. There are two typical approaches: (1) redesigning the projector to directly compress the vision tokens (Yao et al., 2024b; Cha et al., 2024; Chu et al., 2023; 2024; Li et al., 2025; Shen et al., 2024), and (2) pruning unimportant vision tokens (Chen et al., 2024b; Xing et al., 2024; Zhang et al., 2024b; Zhuang et al., 2024). For example, DeCo (Yao et al., 2024b) employs Adaptive Average Pooling, and Qwen2.5-VL (Bai et al., 2025) uses PixelShuffle to compress vision tokens. VisPruner (Zhang et al., 2024a) maintains the domi-

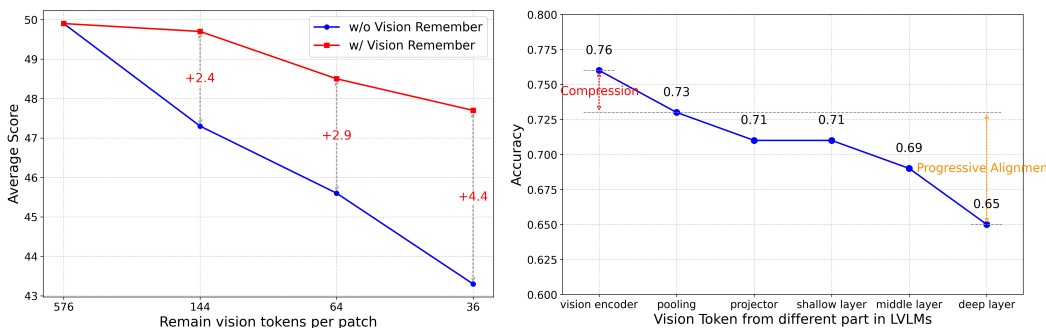

(a) Performance drop with compression ratios.  (b) Linear probing of classification on Tiny-ImageNet

Figure 1: Preliminary analysis. (a) Compressing the vision tokens can cause information loss, resulting in performance degradation. The proposed Vision Remember alleviates this problem. (b) We extract vision tokens from distinct components of LVLM and evaluate the classification accuracy on Tiny-ImageNet. The compression only happens in pooling. Our analysis identifies two primary sources of visual information loss: Information Bottleneck in Token Compression and Visual Cues Forgetting in Progressive Alignment.

nant tokens and prunes the other tokens in the vision encoder. Nonetheless, these methods may lose visual information, which is important for the tasks that rely on fine-grained spatial relationships, such as OCR and Chart & Table understanding.

To systematically identify the reasons for visual information loss, we evaluate the classification capacity of vision tokens extracted from different components of LVLM on the TinyImageNet dataset. Similar to linear probing, we freeze all the parameters in LVLM and only train a lightweight classification head composed of a single cross-attention layer followed by a linear layer. As shown in Fig. 1b, we identify two fundamental reasons for the performance degradation: (1) Information Bottleneck in Token Compression - The compression of vision tokens inevitably discards fine-grained visual details (e.g., texture patterns, small objects), while the surviving tokens lack the representational capacity to reconstruct such high-frequency visual information; (2) Visual Cues Forgetting in Progressive Alignment - During the cross-modal alignment process, where vision tokens sequentially interact with text tokens in the LLM's attention layers, visual features undergo gradual attenuation due to dominant linguistic priors, resulting in visual cues forgetting across the LLM decoder. Hence, we raise a question: *Since the problem comes from the compression in the projector and the forgetting in LLM, can we recover the original vision features between the LVLM decoder layers?*

To answer the above question, this paper present **Vision Remember**, an approach that resamples original visual features multiple times across the LVLM decoder layers to compensate for the lost vision cues. The main motivation is that the features obtained by the vision encoder contain original vision information, and we can re-inject them into vision tokens, not in the projector, but between the decoder layers. Following this principle, we introduce the first key module: **Token-Feature Cross-Attention Layer**, which employs local cross-attention to interact the vision tokens and vision features. Furthermore, we also aggregate multi-level features to enrich the visual representation and enhance the model's ability of visual comprehension. Another key module is **Token Bidirectional Self-Attention Layer**. Casual attention mask inherently restricts cross-token interactions in visual representations while preventing access to subsequent textual cues, consequently disregarding textual descriptions of foreground objects. To address this issue, this module employs self-attention mechanisms to enable mutual attention among vision tokens, and introduces text-guided tokens to implicitly characterize the region of interests.

We evaluate our proposed method on LLaVA-NeXT (Liu et al., 2024c), the most widely adopted baseline in academia, and assess the model's performance through average scores across eleven comprehensive benchmarks. Experimental results demonstrate consistent performance gains when our method is combined with various efficient visual projectors. Specifically, Vision Remember achieves improvements of +3.0 (6.6%), +3.2 (7.2%), and +4.4 (10.1%) for Average Pooling, PixelShuffle, and Perceiver Resampler, respectively. On identical baselines, our approach outperforms prior works, TokenPack (Li et al., 2025) and FastV (Chen et al., 2024b), by margins of +2.7 (5.9%)

and +5.7 (13.3%). To further validate the generalizability of our method, we conduct experiments on two different baselines Qwen2.5-VL (Bai et al., 2025) and MiniCPM-V-2 (Yao et al., 2024c), and observe performance improvements. These experiments demonstrate that Vision Remember can serve as a fundamental component when constructing an efficient LVLM.

## 2 RELATED WORK

### 2.1 LARGE VISION-LANGUAGE MODELS

Many works focus on endowing LLMs with visual understanding capabilities, transforming them into LVLMs (Yao et al., 2024c; Liu et al., 2024e; Chen et al., 2024d; Liu et al., 2024b; Abdin et al., 2024; Liu et al., 2024c; Tong et al., 2025; Wang et al., 2024; Lu et al., 2024). Based on differences in visual signal integration methods, we categorize existing approaches into two classes: (1) Token Concatenation and (2) Visual Feature Sampling.

**Token Concatenation based LVLM.** These methods align the vision tokens into the linguistic domain by a projector, and then concatenate them with text tokens before feeding into the LLM. Blip-2 (Li et al., 2023) and LLaVA (Liu et al., 2024d) both adopt this paradigm, but the main difference is that Blip-2 uses Q-Former to bridge different modalities, while LLaVA directly employs MLP layers to map vision tokens into the language domain. LLaVA-NeXT (Liu et al., 2024c) introduces dynamic image cropping to enhance the fine-grained understanding capabilities. Mini-Gemini (Gao et al., 2024) and Cambrian-1 (Tong et al., 2025) have explored various combination methods of multiple vision encoders. DenseConnector (Yao et al., 2024a) and MMFuser (Cao et al., 2024), enhance existing LVLMs by leveraging multi-level visual features. However, the aforementioned methods primarily focus on enhancing the understanding capabilities of LVLMs, while neglecting the efficiency of the models. Larger foundational models and longer input sequences can result in significant computational resource consumption during inference.

**Visual Feature Sampling based LVLM.** Several approaches inject visual information into LLMs via cross-attention layers, where text tokens serve as queries while visual features act as keys and values. Flamingo (Alayrac et al., 2022) introduced gated x-attention layers, which enable the model to understand visual inputs by employing Recent work has focused on enhancing the visual understanding capabilities of LVLMs. LLaMA 3 (Dubey et al., 2024) also adopts this paradigm, constructing multimodal models with varying parameter counts, and achieves strong performance through large-scale training. EVLM (Chen et al., 2024a) and NVLM (Dai et al., 2024) integrate these two paradigms, constructing hybrid-architecture LVLMs. However, gated cross-attention mechanisms incur significant parameter overhead—for instance, in LLaMA 3, merely 8 cross-attention layers account for 100B parameters. Unlike previous approaches, our method performs sampling exclusively on the vision tokens by leveraging local cross-attention mechanisms. In contrast, prior methods typically employ global attention, which involves sampling across the entire sequence of tokens, including both vision and text modalities. Our design introduces minimal parameters while maintaining model efficiency.

### 2.2 EFFICIENT LARGE VISION LANGUAGE MODELS

Many works focus on improving the efficiency of LVLMs by reducing the number of visual tokens, which can generally be categorized into the following two types: (1) redesigning the projector to directly compress the visual tokens; (2) directly pruning the unimportant vision tokens between the decoder layers.

**Projector Design.** DeCo (Yao et al., 2024b) provides a detailed analysis of the "dual abstraction" phenomenon in Q-Former and proposes using 2D adaptive average pooling directly in the projector to perform downsampling of visual tokens. By utilizing Point-to-Region attention in the local region, TokenPacker (Li et al., 2025) enhances fine-grained understanding capability while preserving spatial information. MobileVLM (Chu et al., 2023; 2024) introduces a convolutional LDP module for visual token compression, whereas Qwen2-VL (Wang et al., 2024) and InternVL (Chen et al., 2024e) employ PixelShuffle.

**Vision Token Pruning.** FastV (Chen et al., 2024b) introduces a method that prunes the last top-k visual tokens based on attention values. This plug-and-play approach can be integrated into vari-

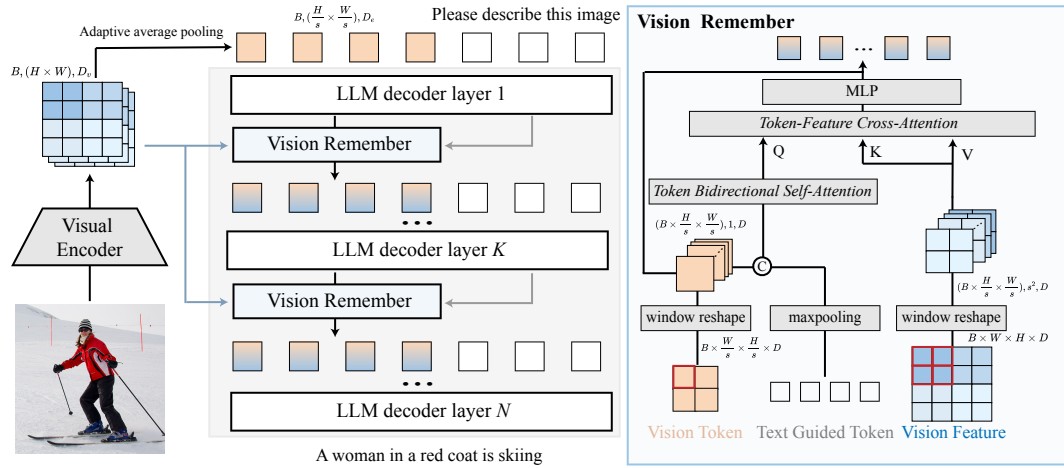

Figure 2: **Vision Remember framework.** We insert the proposed Vision Remember between the LLM decoder layers. Adaptive Average Pooling is used to compress the vision tokens. In Vision Remember, we adopt local attention as shown in the blue part. A vision token only focuses on a $s \times s$ local region in the multi-level vision feature to improve the computational efficiency and capture the fine-grained spatial information.

ous LVLMs in a training-free paradigm. SparseVLM (Zhang et al., 2024b) proposed a rank-based strategy to adaptively determine the pruning ratio for each layer. PyramidDrop (Xing et al., 2024) divides the entire LVLM decoder into multiple stages and performs pruning at a fixed ratio after the last layer in each stage. VisionZip (Yang et al., 2025) and VisPruner (Zhang et al., 2024a) prunes the redundant tokens in the vision encoder, while preserving the structural integrity of LLM.

Unlike existing approaches, our proposed method focuses on recovering lost visual cues rather than further optimizing model operational efficiency.

## 3 VISION REMEMBER

In this section, we first give a brief introduction to the widely used LLaVA series (Liu et al., 2024c; Li et al., 2024a; Liu et al., 2024b;d), which serve as our baseline. We then introduce our proposed Vision Remember, including two key components: Token-Feature Cross-Attention Layer and Token Bidirectional Self-Attention Layer. Notably, Vision Remember is not only bound to LLaVA but also can be integrated into other Efficient LVLMs.

### 3.1 PRELIMINARY

We choose the widely used LLaVA-NeXT (Liu et al., 2024c) as our baseline, which consists of three components: 1) Vision Encoder, 2) Vision Projector, and 3) Large Language Model. Vision Encoder, typically a Vision Transformer (ViT) or Convolution Neural Network (CNN) that has been trained on a large amount of data, is primarily used to extract vision features from the input image. Then, a 2-layer MLP named Vision Projector is adopted to align the vision features with linguistic space. Finally, the text tokens $\mathbf{T}_t$ and the vision tokens $\mathbf{T}_v$ after alignment are concatenated and fed into a LLM to generate the response $\mathbf{R}$ with length $L$ in an auto-regressive manner, which can be formulated as:

$$p(\mathbf{R}|\mathbf{T}_v, \mathbf{T}_t) = \prod_{l=1}^{L} p(r_l|\mathbf{T}_v, \mathbf{T}_t, r_{<l}). \tag{1}$$

### 3.2 TOKEN-FEATURE CROSS-ATTENTION LAYER

To compensate for the lost visual information and recover the visual cues, we retain the original vision feature and interact with the vision tokens from the LLM decoder layers.

**Multi-level Vision Feature.** Many studies have demonstrated that different layers in ViT (Dosovitskiy et al., 2020) exhibit different attention patterns. Shallow layers tend to focus on low-level local spatial information, while deeper layers tend to emphasize global semantic features. Effectively utilizing the multi-level vision features can significantly enhance the LVLM's parsing and understanding capability. Here, we directly concatenate the vision features from different layers along the feature dimension to form information-rich vision features.

**Local Attention** To avoid disrupting the inductive bias in images and better utilize spatial structural information, we adopt the local attention mechanism during the interaction process. As shown in Fig.2, given the vision token from the LLM decoder layers, we first expand its dimension by a MLP layer to match the vision features dimension $D$. Then following Swin Transformer (Liu et al., 2021), we divide the vision features into $n^2$ local regions of size $s \times s$ in the spatial dimension $H \times W$ (typically, $H = W$), where $n = H/s$. We reshape the partitioned vision features $F_v \in \mathbb{R}^{B \times H \times W \times D}$ to $\mathbb{R}^{(B \cdot n^2) \times s^2 \times D}$, to serve as key and value. For vision tokens, we perform the same operation, but the size of the local region is $1 \times 1$, serving as the query. In this way, a vision token only performs cross-attention with an $s \times s$ local region, rather than attending globally to all vision features.

The benefits of using local attention can be summarized as follows: (1) computational efficiency and (2) localized contextual information. First, local attention reduces the computational complexity compared to traditional global attention mechanisms. By partitioning the vision features into smaller local regions, each vision token only attends to a limited number of visual features, resulting in faster processing and improved efficiency. Second, local attention allows each vision token to focus on a specific local region of vision features. This attention mechanism helps capture more fine-grained contextual information and spatial relationships within the region, leading to better understanding and representation of the visual content.

**Other choices.** There are three optional interaction mechanisms in Vision Remember: (1) Local Attention, (2) Deformable Cross Attention (Zhu et al., 2020; Shen et al., 2024), and (3) Naive Global Attention (Vaswani et al., 2017). For Deformable Cross Attention and Naive Global Attention, we all use vision tokens as query, vision features as key & value, but the difference is that the former uses deformable attention to deal with multi-level vision features and enhance sparse spatial information. The comparison between the three interaction mechanisms in Sec.4.4 shows that Local Attention and Deformable Cross Attention both get positive promotion, and the former gets the best performance.

### 3.3 TOKEN BIDIRECTIONAL SELF-ATTENTION LAYER

As mentioned in Sec.1, causal attention mask used in the LLM decoder ensures each token can only attend to preceding tokens in the sequences. This is naturally suited for language modeling, as textual signals are inherently sequential. However, visual signals are inherently two-dimensional and encode rich spatial relationships. Imposing causal masking during visual token modeling would fundamentally restrict cross-token interactions in visual representations (Liu et al., 2024f; Zhu et al., 2024; Li et al., 2024b). To address this problem, we introduce Token Bidirectional Self-Attention Layer, which employs self-attention mechanism with full attention.

In current Large Vision-Language Models (LVLMs) such as LLaVA and Qwen2.5-VL, visual tokens are typically prepended to text tokens during sequence concatenation. The inherent property of causal attention prevents visual tokens from perceiving subsequent text tokens, effectively causing the model to disregard user prompt inputs when processing visual tokens. User prompts often contain referential attributes for target objects or foreground elements (e.g., 'the person wearing red' versus 'the person wearing blue'). Ignoring such input priors prevents the model from distinguishing which visual tokens actually merit attention (i.e., those containing the referenced foreground) during visual token processing. We first extract text tokens from hidden states in the decoder layer, and then compress them along the sequence dimension with Adaptive Max Pooling to get the text-guided token. Finally, we concatenate vision tokens with the text-guided token to enable fully cross-modal interaction through the self-attention mechanism.

### 3.4 TRAINING.

Following the common practice, we train the Vision Remember in multiple phases.

Table 1: **Performance gain with various efficient vision projectors.** Performance with proposed *Vision Remember* is marked in gray . *A.A.P* means *Adaptive Average Pooling*. Our proposed method can improve the LVLM's ability of visual parsing and understanding when combined with various efficient vision projectors.

| Projectors | GQA | $MME^P$ | RWQA | $SQA^I$ | AI2D | $MMMU^V$ | MMStar | ChartQA | DocVQA | TextVQA | OCRBench | Avg. |
|---|---|---|---|---|---|---|---|---|---|---|---|---|
| A.A.P | 57.4 | 1214.8 | 47.6 | 54.8 | 53.1 | 30.1 | 35.6 | 36.5 | 52.3 | 41.0 | 31.8 | **45.5** |
| | 58.7 | 1176.2 | 51.9 | 55.9 | 51.6 | 29.6 | 37.1 | 48.7 | 54.2 | 49.6 | 37.3 | **48.5**(+3.0) |
| PixelShuffle | 56.8 | 1234.7 | 47.6 | 56.3 | 52.3 | 30.7 | 37.1 | 32.2 | 47.8 | 39.4 | 30.3 | **44.8** |
| | 58.8 | 1144.1 | 50.7 | 54.3 | 52.8 | 29.1 | 37.4 | 48.3 | 53.3 | 48.8 | 36.8 | **48.0**(+3.2) |
| Perceiver | 55.7 | 1213.5 | 45.9 | 57.2 | 52.6 | 30.7 | 36.3 | 31.3 | 39.5 | 42.1 | 28.8 | **43.7** |
| | 59.0 | 1149.2 | 50.1 | 55.7 | 52.2 | 29.3 | 36.8 | 48.7 | 53.8 | 49.3 | 37.3 | **48.1**(+4.4) |
| LDPv2 | 57.8 | 1224.8 | 48.2 | 54.7 | 52.6 | 30.3 | 34.8 | 39.5 | 53.4 | 43.4 | 32.6 | **46.2** |
| | 58.7 | 1191.1 | 50.1 | 55.9 | 51.8 | 29.5 | 37.3 | 47.5 | 53.9 | 49.3 | 36.9 | **48.2**(+2.0) |

**Phase-1: Language-Image Alignment.** In this phase, we use the image-caption pairs in the CC-558K dataset (Liu et al., 2024d) to train the Vision Projector and Vision Remember, keeping the Vision Encoder and LLM frozen. The main purpose of this phase is to align the hidden representation space between the vision and language modalities.

**Phase-2: Visual Instruct Tuning.** In this phase, we include the LLM in training. The 779K mixture dataset (Liu et al., 2024c) is used to enhance the LVLM's ability of vision understanding and instruction following. To support high-resolution input images, the AnyRes (Li et al., 2024a) technique is adopted during this phase.

# 4 EXPERIMENT

## 4.1 IMPLEMENTATION DETAILS

We choose the widely used LLaVA-NeXT (Liu et al., 2024c) as baseline, SigLIP-Large (Zhai et al., 2023) as Vision Encoder and Qwen2 (Yang et al., 2024) series as LLM. The size of per tile of input image is resized to $384 \times 384$, so the shape of feature map from SigLIP-Large-patch16-384 is $24 \times 24$, and then 2D Adaptive Average Pooling is employed to compress the spatial resolution to $8 \times 8$, resulting in 64 vision tokens per patch (*i.e.* compression ratio is 1/9). If not specified, we select layers 7, 15, and 23 from the vision encoder to form multi-level vision features and insert Vision Remember before the first and fourth decoder layers. We train all models for one epoch, and use the AdamW optimizer with Cosine learning rate schedule. In phase-1, the learning rate is 1e-3 and the batch size is 256, and in phase-2, the learning rate is 2e-4 and the batch size is 32. The experiments are conducted on $8 \times$ Nvidia A100 GPUs.

## 4.2 BENCHMARKS

We conduct extensive experiments on 11 benchmarks to validate the understanding and parsing capabilities of the proposed method. The benchmarks can be divided into the following types based on different focus areas: (1) General Question Answer benchmarks include GQA (Hudson & Manning, 2019), MME-Perception (Fu et al., 2024) and RealWorldQA (xAI team, 2024). (2) Comprehensive Knowledge Reasoning benchmarks include ScienceQA_Image (Lu et al., 2022), AI2D (Kembhavi et al., 2016), MMMU (Yue et al., 2024) and MMStar (Chen et al., 2024c). (3) OCR&Chart Parsing benchmarks include ChartQA (Masry et al., 2022), DocVQA (Mathew et al., 2021), TextVQA (Singh et al., 2019) and OCRBench (Liu et al., 2023). To compare the performance of LVLM, we take the average scores on the whole benchmark.

## 4.3 MAIN RESULTS

**Performance gain with various efficient vision projectors.** To demonstrate the effectiveness of

Table 2: **Performance gain with various compression ratio.** Performance with proposed ***Vision Remember*** is marked in `gray`. *Adaptive Average Pooling* is used in projector to downsample the vision tokens. The proposed method demonstrates consistent performance improvements across varying compression ratios, with greater performance gains observed at higher compression ratios (*i.e.*, fewer retained tokens).

| Comp. Ratio | GQA | $MME^p$ | RWQA | $SQA^I$ | AI2D | $MMMU^v$ | MMStar | ChartQA | DocVQA | TextVQA | OCRBench | Avg. |
|---|---|---|---|---|---|---|---|---|---|---|---|---|
| 4 | 57.5 | 1174.7 | 48.0 | 55.8 | 53.1 | 28.9 | 36.7 | 43.6 | 56.5 | 46.2 | 35.0 | **47.3** |
|  | 59.5 | 1205.6 | 49.7 | 54.7 | 53.5 | 30.7 | 37.5 | 52.1 | 56.4 | 51.7 | 40.3 | **49.7**(+2.4) |
| 9 | 57.4 | 1214.8 | 47.6 | 54.8 | 53.1 | 30.1 | 35.6 | 36.5 | 52.3 | 41.0 | 31.8 | **45.5** |
|  | 58.7 | 1176.2 | 51.9 | 55.9 | 51.6 | 29.6 | 37.1 | 48.7 | 54.2 | 49.6 | 37.3 | **48.5**(+3.0) |
| 16 | 56.7 | 1181.8 | 45.8 | 54.5 | 51.9 | 31.8 | 36.5 | 29.6 | 47.4 | 36.7 | 26.9 | **43.3** |
|  | 58.4 | 1207.0 | 48.8 | 56.9 | 52.9 | 31.2 | 35.5 | 45.5 | 52.6 | 46.9 | 35.7 | **47.7**(+4.4) |

Vision Remember, we report the performance when combined with various efficient vision projectors (Yao et al., 2024b; Shen et al., 2024; Chen et al., 2024d; Chu et al., 2023; 2024). Just as Tab.1 shows, when different projectors are combined with Vision Remember, the LVLM's ability of visual understanding are all improved. Specifically, the proposed method can lift the average score of *Adaptive Average Pooling* by +3.0, *PixelShuffle* by +3.2, and *Perceiver Resamplers* by +4.4. The higher improvements are primarily concentrated on benchmarks including GQA, RealWorldQA, ChartQA, DocVQA, TextVQA, and OCRBench, which demonstrates that Vision Remember can alleviate the visual information loss and enhance the LVLM's ability to understand fine-grained visual features and spatial relationships, especially in tasks such as OCR and Chart/Table analysis.

**Performance gain with various compression ratios.** Tab.2 presents the performance gains with various compression ratios. We first employ the Adaptive Average Pooling to compress the vision tokens with three ratios: 4×, 9×, and 16×, *i.e* 144, 64 and 36 vision tokens remain in each patch, respectively. Then we integrate the proposed Vision Remember and compare the average score on 11 benchmarks. Specifically, our method achieves performance gains of +2.4, +3.0, and +4.1 at compression ratios of 4×, 9×, and 16×, respectively. These results demonstrate that Vision Remember consistently improves performance across varying compression rates, with greater performance gains observed at higher compression ratios.

**Comparison with other efficient methods.** Tab.3 presents the performance comparison with other efficient methods, including the pruning-based methods FastV (Chen et al., 2024b), PyramidDrop (Xing et al., 2024), VisPruner (Zhang et al., 2024a), and compress-based methods DeCo (Yao et al., 2024b), TokenPacker (Li et al., 2025). For fair comparison, we keep the experiment under consistent settings, including the training data, model size, and compression ratio. Our approach achieves the best average accuracy across all three LLM scales in Tab.3: 48.5 with Qwen2-0.5B (+1.9 over VisPruner and +3.3 over TokenPacker), 55.5 with Qwen2-1.5B (+1.4 over VisPruner and +3.0 over TokenPacker), and 60.2 with Qwen2-7B (+1.6 over VisPruner and +2.9 over TokenPacker). Notably, VisPruner prunes redundant visual tokens in the vision encoder, while FastV and PyramidDrop perform token pruning within the LLM. All these methods rely on attention maps to determine which tokens to retain or drop. However, Flash Attention (Dao et al., 2022) and Scaled Dot-Product Attention (SDPA)—widely adopted techniques for accelerating attention computation—do not support the output of attention maps by design. Consequently, the aforementioned pruning methods cannot be fully integrated with these accelerating techniques at certain layers, leading to significant efficiency bottlenecks. We will provide a detailed comparative analysis in Sec.4.5. Compared with DeCo and TokenPacker, our method not only consider the information bottleneck in token compression, but also recover the lost visual cues in progressive alignment, thus achieves better performance.

### 4.4 ABALTION STUDY

**Key Components.** Tab.4a presents the results of the ablation study that evaluate the contributions of different key components. By incrementally adding Local Attention, Multi-level Fusion, Bidirectional Interaction, and Text guided Token, the results demonstrate performance gains over the

Table 3: **Performance comparison with other efficient methods.** We reproduce these methods under the consistent settings. Blue means performance drop compared with our method.

| Method | GQA | MME$^P$ | RWQA | SQA$^I$ | AI2D | MMMU$^V$ | MMStar | ChartQA | DocVQA | TextVQA | OCRBench | Avg. |
|---|---|---|---|---|---|---|---|---|---|---|---|---|
| *Qwen2-0.5B as LLM* | | | | | | | | | | | | |
| FastV (Chen et al., 2024b) | 55.1 | 1141.4 | 48.1 | 55.4 | 50.8 | 29.6 | 34.4 | 25.6 | 42.4 | 44.7 | 27.9 | **42.8(-5.7)** |
| PDrop (Xing et al., 2024) | 55.1 | 1204.8 | 50.9 | 57.5 | 52.5 | 30.2 | 35.4 | 35.3 | 47.6 | 47.8 | 25.7 | **45.3(-3.2)** |
| VisPruner (Zhang et al., 2024a) | 57.6 | 1194.2 | 49.3 | 56.9 | 53.4 | 29.3 | 36.1 | 38.8 | 43.7 | 50.2 | 38.0 | **46.6(-1.9)** |
| DeCo (Yao et al., 2024b) | 57.4 | 1214.8 | 47.6 | 54.8 | 53.1 | 30.1 | 35.6 | 36.5 | 52.3 | 41.0 | 31.8 | **45.5(-3.0)** |
| TokenPacker (Li et al., 2025) | 57.3 | 1175.8 | 50.3 | 55.3 | 51.9 | 31.7 | 36.6 | 38.8 | 50.7 | 42.9 | 30.1 | **45.8(-2.7)** |
| Ours | 58.7 | 1176.2 | 51.9 | 55.9 | 51.6 | 29.6 | 37.1 | 48.7 | 54.2 | 49.6 | 37.3 | **48.5** |
| *Qwen2-1.5B as LLM* | | | | | | | | | | | | |
| FastV (Chen et al., 2024b) | 57.2 | 1329.1 | 53.3 | 70.0 | 59.7 | 33.3 | 39.8 | 41.5 | 48.0 | 49.8 | 29.8 | **50.0(-5.5)** |
| PDrop (Xing et al., 2024) | 56.3 | 1324.6 | 56.0 | 69.5 | 60.6 | 33.1 | 40.0 | 46.2 | 62.9 | 55.1 | 25.7 | **52.0(-3.5)** |
| VisPruner (Zhang et al., 2024a) | 59.8 | 1323.3 | 54.5 | 69.2 | 62.8 | 32.2 | 39.6 | 48.4 | 62.9 | 57.8 | 42.3 | **54.1(-1.4)** |
| DeCo (Yao et al., 2024b) | 61.3 | 1338.9 | 52.4 | 68.7 | 62.8 | 34.8 | 38.6 | 47.6 | 63.4 | 49.4 | 38.9 | **53.2(-2.3)** |
| TokenPacker (Li et al., 2025) | 60.3 | 1361.6 | 54.0 | 67.6 | 62.3 | 33.6 | 37.7 | 46.1 | 61.8 | 50.3 | 36.2 | **52.5(-3.0)** |
| Ours | 62.6 | 1360.6 | 57.7 | 68.2 | 63.7 | 32.7 | 38.8 | 54.8 | 63.4 | 56.9 | 44.1 | **55.5** |
| *Qwen2-7B as LLM* | | | | | | | | | | | | |
| FastV (Chen et al., 2024b) | 59.7 | 1474.7 | 59.2 | 67.3 | 68.1 | 35.8 | 43.0 | 51.2 | 57.2 | 54.1 | 34.5 | **54.9(-5.3)** |
| PDrop (Xing et al., 2024) | 58.9 | 1467.6 | 60.3 | 70.3 | 66.5 | 38.0 | 41.5 | 50.6 | 67.8 | 61.4 | 32.8 | **56.5(-3.7)** |
| VisPruner (Zhang et al., 2024a) | 62.1 | 1474.1 | 58.0 | 69.1 | 70.2 | 35.7 | 43.1 | 60.0 | 65.5 | 61.7 | 42.3 | **58.3(-1.9)** |
| DeCo (Yao et al., 2024b) | 61.6 | 1425.4 | 58.7 | 70.8 | 70.4 | 38.2 | 44.4 | 58.8 | 67.9 | 56.3 | 42.6 | **58.2(-2.0)** |
| TokenPacker (Li et al., 2025) | 60.6 | 1463.6 | 58.3 | 73.5 | 69.2 | 36.9 | 43.5 | 56.2 | 66.3 | 53.6 | 39.2 | **57.3(-2.9)** |
| Ours | 62.2 | 1488.9 | 62.0 | 73.0 | 71.4 | 38.6 | 44.5 | 62.0 | 68.1 | 60.7 | 44.9 | **60.2** |

baseline. The baseline achieves an average score of 42.9, whereas Local Attention boosts the average to 45.7. Introducing Multi-level Fusion further increases the average to 46.3, and integrating Bidirectional Interaction achieves an average score of 46.6. Notably, the simultaneous use of all yields the highest average score of 46.7, an improvement of +3.8 over the baseline. The results clearly indicate that each module positively contributes to overall performance, and their combined usage provides the most significant enhancement, particularly in complex tasks such as OCR&Chart understanding.

**Interaction Methods in Vision Remember.** Tab.4b investigates the impact of different interaction methods within the Vision Remember. Due to focus on all image features without taking into account the visual local context information, Global Attention yields the poorest results (average score 43.7). Deformable Attention takes into account local sparse sampling, but learning the offsets can cause the model confusion about reasonable sampling points, leading to suboptimal result (average score 45.1). Local Attention achieves the best results (average score 46.7).

**Insertion Position in LLM.** We also conduct ablation study on the insertion layers of Vision Remember, and the results are reported in the Tab.4c. If we insert vision remember after the first layer, the average score gets +3.4 improvement and yields 46.3 When we insert vision remember after the first and fourth layers, the average score yields 46.7. Further insertion into subsequent layers leads to the performance saturating without measurable gains in the average metric. This is because the middle layers are 'thinking and reasoning' (Wu et al., 2024; Yu & Lee, 2025; Basu et al., 2024), and introducing too many visual features may destroy this pattern.

## 4.5 MORE ANALYSIS

**Performance gain on other baselines.** We also evaluate the proposed method on two different baselines: Qwen2.5-VL-3B (Bai et al., 2025) and MiniCPM-V-3B (Yao et al., 2024c). Qwen2.5-VL employs NaViT (Dehghani et al., 2023) as vision encoder and pixelshuffle merger to compress the vision tokens, while MiniCPM-V uses MiniCPM as LLM and Q-Former like perceiver resampler as

Table 4: Ablation studies. The default setting is marked in  gray .

(a) Ablation study of key components.

| | Local Attn | Multi-level | Bidir Interact | Text Token | General | Knowledge | OCR&Chart | Average Score |
|---|---|---|---|---|---|---|---|---|
| Baseline | | | | | 54.6 | 43.7 | 33.4 | **42.9** |
| Vision Remember | ✓ | | | | 56.4 | 43.7 | 39.5 | **45.7**(+2.8) |
| | ✓ | ✓ | | | 55.4 | 44.0 | 42.2 | **46.3**(+3.4) |
| | ✓ | ✓ | ✓ | | 55.7 | 43.9 | 42.4 | **46.6**(+3.7) |
| | ✓ | ✓ | ✓ | ✓ | 55.9 | 43.9 | 42.5 | **46.7**(+3.8) |

(b) Experimental results with various **interaction methods** in Vision Remember.

| Interaction | General | Knowledge | OCR&Chart | Average Score |
|---|---|---|---|---|
| Global Attn | 54.1 | 43.3 | 36.4 | **43.7** |
| Defor Attn | 54.8 | 43.0 | 40.0 | **45.1** |
| Local Attn | 55.9 | 43.9 | 42.5 | **46.7** |

(c) Experimental results with various **insertion positions** of Vision Remember.

| Insertion | General | Knowledge | OCR&Chart | Average Score |
|---|---|---|---|---|
| 1 | 55.5 | 43.6 | 42.0 | **46.3** |
| 1,4 | 55.9 | 43.9 | 42.5 | **46.7** |
| 1,4,7 | 55.9 | 44.0 | 42.5 | **46.7** |

Table 5: More analysis on various baselines and efficiency. The default setting is marked in  gray .

(a) Performance gain on different baselines.

| Baseline | General | Knowledge | OCR&Chart | Average Score |
|---|---|---|---|---|
| Qwen2.5-VL | 58.1 | 51.2 | 59.5 | **56.1** |
| | 59.7 | 53.2 | 60.9 | **57.8**(+1.7) |
| MiniCPM-V | 56.6 | 57.4 | 40.9 | **50.4** |
| | 58.1 | 58.1 | 41.5 | **51.5**(+1.1) |

(b) Efficiency comparison on a NVIDIA A100 GPU.

| Methods | TTFT/ms ↓ | TPS ↑ | Average Score |
|---|---|---|---|
| LLaVA-NeXT | 150.9 | 35.9 | **49.8** |
| VisPruner | 151.3 | 44.8 | **47.5** |
| TokenPacker | 103.8 | 45.2 | **47.6** |
| Ours | 104.2 | 45.1 | **49.7** |

projector. Since none of them released their training data, we re-trained the models on the LLaVA-NeXT (Liu et al., 2024c) training set, and the final results are reported in Tab.5a. Our proposed method achieves performance improvements across two baselines, proving its effectiveness and robustness. This experiment also demonstrates that Vision Remember could be considered as a basic component when constructing an Efficient LVLM.

**Efficiency analysis.** Tab.5b presents the efficiency comparison. We chose two metrics, TTFT (Time to First Token) and TPS (Tokens per Second), to evaluate the efficiency of the proposed method and others. TTFT reflects the prefilling stage (limited on computational capacity) latency, and TPS indicates the decoding stage (limited on memory bandwidth) efficiency. Compared with LLaVA-NeXT (Liu et al., 2024c), which does not compress the vision tokens, our method saves 46.7ms (31%) in the prefilling stage, and improves TPS to 45.1, while only gets 0.1 (0.2%) drop on the average score. Although the Vision Remember module remains inactive during the decoding stage, our method reduces the KV cache length (because of the vision token compression in the prefilling stage) and improves the decoding efficiency. VisPruner relies on attention maps to select important tokens and could not be compatible with Flash Attention or SDPA. Consequently, it cannot accelerate the compute-bound prefilling phase.

## 5 CONCLUSION

In this paper, we investigate the visual information loss in Efficient LVLMs and identify two reasons: Information Bottleneck and Visual Cues Forgetting. And then we propose Vision Remember to recover the lost visual information with vision feature resampling. Equipped with Token-Feature Cross-Attention Layer and Token Bidirectional Self-Attention Layer, the proposed method captures more fine-grained contextual information and spatial relationships, enhancing the capability of visual parsing and understanding. Comprehensive experiments validate the effectiveness of the proposed method when combined with various efficient vision projectors and LVLMs. We hope our work can promote community interest in Efficient LVLMs, especially on small models with fewer parameters.

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

# A  APPENDIX

## A.1  USE OF LARGE LANGUAGE MODELS

We used the Large Language Model only for language refinement of this manuscript, including grammar, clarity, and readability improvements. No technical content, experimental design, analysis, or conclusions were generated or influenced by the LLM. All scientific ideas, methods, and results are solely the authors' original work.

## A.2  MORE EXPERIMENTS

Due to page limitations, we give more experimental analysis in the Appendix.

Table 6: Performance comparison with SVA aggregation. Our method is marked in gray.

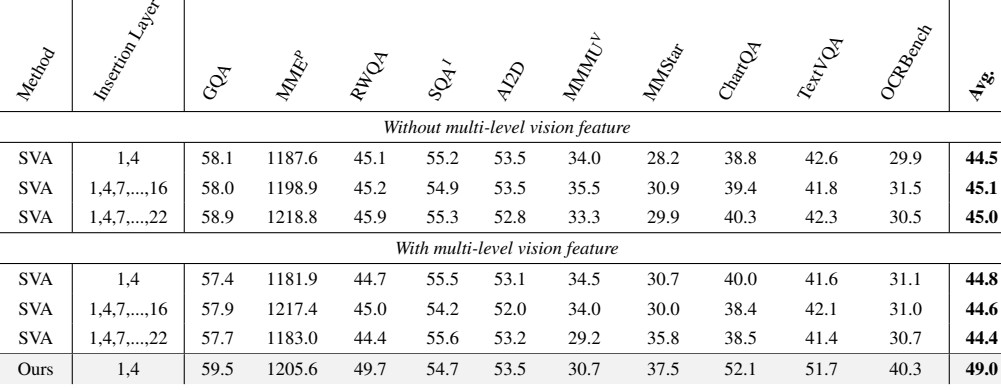

| Method | Insertion Layer | GQA | MME$^p$ | RWQA | SQA$^I$ | AI2D | MMMU$^v$ | MMStar | ChartQA | TextVQA | OCRBench | Avg. |
|---|---|---|---|---|---|---|---|---|---|---|---|---|
| *Without multi-level vision feature* | | | | | | | | | | | | |
| SVA | 1,4 | 58.1 | 1187.6 | 45.1 | 55.2 | 53.5 | 34.0 | 28.2 | 38.8 | 42.6 | 29.9 | **44.5** |
| SVA | 1,4,7,...,16 | 58.0 | 1198.9 | 45.2 | 54.9 | 53.5 | 35.5 | 30.9 | 39.4 | 41.8 | 31.5 | **45.1** |
| SVA | 1,4,7,...,22 | 58.9 | 1218.8 | 45.9 | 55.3 | 52.8 | 33.3 | 29.9 | 40.3 | 42.3 | 30.5 | **45.0** |
| *With multi-level vision feature* | | | | | | | | | | | | |
| SVA | 1,4 | 57.4 | 1181.9 | 44.7 | 55.5 | 53.1 | 34.5 | 30.7 | 40.0 | 41.6 | 31.1 | **44.8** |
| SVA | 1,4,7,...,16 | 57.9 | 1217.4 | 45.0 | 54.2 | 52.0 | 34.0 | 30.0 | 38.4 | 42.1 | 31.0 | **44.6** |
| SVA | 1,4,7,...,22 | 57.7 | 1183.0 | 44.4 | 55.6 | 53.2 | 29.2 | 35.8 | 38.5 | 41.4 | 30.7 | **44.4** |
| Ours | 1,4 | 59.5 | 1205.6 | 49.7 | 54.7 | 53.5 | 30.7 | 37.5 | 52.1 | 51.7 | 40.3 | **49.0** |

**Comparison with SVA aggregation.** We compare the proposed Vision Remember with SVA Aggregation(Tong et al., 2025), and the results are shown in Tab.6. Since SVA emphasizes the multiple vision encoders ensemble, different from the starting point of efficient LVLM, we extracted the aggregation method separately and integrated it into our baseline model. Specifically, we retain only a single vision encoder and compress the vision tokens with average pooling in the vision projector. In the aggregation phase, we also incorporated multi-layer vision features. It can be observed that SVA Aggregation does not utilize multi-layer vision features effectively, and its performance is also lower than our method.

**Multi-level Vision Features in Vision Remember.** Vision Remember effectively utilizes multi-level vision features. We have conducted ablation experiments on this key component, and the results are shown in the Tab.7. When only the visual features from the 23rd layer (the same as the Vision Projector) are used, the average score is 45.4, compared to the baseline (44.4, which can be calculated from the Tab.1), showing an improvement of +1.0. We can observe that as the number of sampled layers increases, the average score gradually improves. To accelerate the experiments, we use 3 layers fusion in validation experiments.

Table 7: Ablation results on vision features from different SigLip layers

| Layers | GQA | MME$^P$ | RWQA | SQA$^I$ | AI2D | MMMU$^v$ | MMStar | ChartQA | DocVQA | TextVQA | OCRBench | Avg. |
|---|---|---|---|---|---|---|---|---|---|---|---|---|
| 23 | 58.1 | 1239.7 | 49.2 | 54.0 | 52.4 | 31.9 | 36.6 | 41.2 | 41.0 | 44.6 | 31.3 | **45.7** |
| 11-23 | 58.8 | 1191.3 | 47.7 | 55.0 | 52.4 | 31.6 | 36.5 | 44.3 | 43.6 | 46.0 | 32.5 | **46.2** |
| 7-15-23 | 59.5 | 1209.9 | 47.7 | 55.7 | 52.9 | 29.9 | 37.1 | 44.1 | 45.0 | 47.3 | 33.6 | **46.7** |
| 5-11-17-23 | 58.8 | 1184.5 | 47.5 | 55.8 | 53.5 | 29.9 | 36.5 | 46.5 | 44.8 | 48.6 | 33.7 | **46.7** |

## A.3 DETAILED ABLATION RESULTS

Due to page limitations, we give detailed ablation results (as the same as in Sec.4.4) in the Appendix.
**Performance gain on other baselines.**

Table 8: Performance gain on different baselines.

| Baseline | GQA | MME$^P$ | RWQA | SQA$^I$ | AI2D | MMMU$^v$ | MMStar | ChartQA | DocVQA | TextVQA | OCRBench | Avg. |
|---|---|---|---|---|---|---|---|---|---|---|---|---|
| MiniCPM-V-2 | 50.3 | 1348.9 | 52.3 | 74.5 | 58.3 | - | 39.4 | 41.1 | 33.6 | 52.1 | 36.8 | **50.4** |
|  | 51.3 | 1404.2 | 52.8 | 75.2 | 59.1 | - | 39.9 | 41.6 | 33.6 | 52.2 | 38.6 | **51.5(+1.1)** |
| Qwen2.5-VL | 59.7 | 1347.4 | 47.3 | 59.0 | 67.0 | 34.6 | 44.2 | 61.8 | 61.1 | 62.6 | 52.6 | **56.1** |
|  | 59.9 | 1438.1 | 47.3 | 63.5 | 67.6 | 36.2 | 45.5 | 62.6 | 61.4 | 63.0 | 56.5 | **57.8(+1.7)** |

**Insertion Position in LLM.**

Table 9: Abalation on insertion position.

| Comp. Ratio | GQA | MME$^P$ | RWQA | SQA$^I$ | AI2D | MMMU$^v$ | MMStar | ChartQA | DocVQA | TextVQA | OCRBench | Avg. |
|---|---|---|---|---|---|---|---|---|---|---|---|---|
| 1 | 58.9 | 1197.6 | 47.8 | 56.2 | 53.1 | 29.2 | 35.9 | 45.3 | 44.3 | 47.2 | 31.3 | **46.3** |
| 1,4 | 59.5 | 1209.9 | 47.7 | 55.7 | 52.9 | 29.9 | 37.1 | 44.1 | 45.0 | 47.3 | 33.6 | **46.7** |
| 1,4,7 | 58.8 | 1228.0 | 47.5 | 56.5 | 52.7 | 30.2 | 36.7 | 45.5 | 43.7 | 47.3 | 33.5 | **46.7** |
| 1,4,7,10 | 58.9 | 1176.9 | 48.1 | 54.5 | 52.8 | 29.1 | 38.8 | 44.1 | 44.4 | 47.8 | 34.2 | **46.5** |

**Interaction Methods in Vision Remember.**

Table 10: Ablation on interaction methods in Vision Remember..

| Interaction | GQA | MME$^P$ | RWQA | SQA$^I$ | AI2D | MMMU$^v$ | MMStar | ChartQA | DocVQA | TextVQA | OCRBench | Avg. |
|---|---|---|---|---|---|---|---|---|---|---|---|---|
| Global Attn | 56.3 | 1199.8 | 45.9 | 54.2 | 52.8 | 30.8 | 35.5 | 36.4 | 37.7 | 41.4 | 29.9 | **43.7** |
| Deformable Attn | 57.5 | 1234.4 | 45.2 | 54.5 | 52.1 | 30.3 | 35.0 | 42.7 | 41.6 | 43.9 | 31.6 | **45.1** |
| Local Attn | 59.5 | 1209.9 | 47.7 | 55.7 | 52.9 | 29.9 | 37.1 | 44.1 | 45.0 | 47.3 | 33.6 | **46.7** |

## A.4 REPRODUCIBILITY STATEMENT

We utilize 558K image-caption pairs from the LLaVA-filtered CC3M dataset `https://huggingface.co/datasets/liuhaotian/LLaVA-Pretrain` for pretraining and 779K mixture instruction following data `https://huggingface.co/datasets/lmms-lab/LLaVA-NeXT-Data` for instruction tuning, which are all publicly and freely available for academic research. We also use LLaVA-OneVison-SI datasets `https://huggingface.co/`

`datasets/lmms-lab/LLaVA-OneVision-Data`, and this dataset is also publicly and freely available for academic research. We implement all methods with LLaVA-NeXT (`https://github.com/LLaVA-VL/LLaVA-NeXT` codebase, which are released under the Apache-2.0 license.

