# OpenReview forum: "Vision Remember: Recovering Visual Information in Efficient LVLM with Vision Feature Resampling"
_ICLR.cc/2026/Conference — ICLR 2026 Conference Withdrawn Submission_

### Official Review · Reviewer_TWmZ · 2025-10-26

**Soundness:** 3
**Presentation:** 3
**Contribution:** 3
**Rating:** 4
**Confidence:** 5

**Summary:**

The paper discusses the accuracy-efficiency challenge in vision-language models, where feeding too many visual tokens to the large decoder model sacrifices efficiency, and using existing token pruning and merging methods sacrifices accuracy, particularly for tasks requiring fine-grained details such as OCR. The paper proposes a module called “Vision Remember,” which is added to a few layers of the decoder architecture to help the model access fine-grained visual features without increasing the number of tokens. Using a combination of open-weight vision encoders, language models, and public datasets, the paper shows several results supporting the empirical benefits of the proposed method.

**Strengths:**

- The research direction discussed in the paper (efficiency consequences of augmenting a large decoder model with visual tokens) is valuable and important for practical use cases.
- Paper presentation is generally good and easy to follow.
- The proposed method is intuitive: it allows visual tokens to access original visual features to tackle the information bottleneck caused by visual token compression.
- The preliminary linear probing results shown in Figure 1 are interesting.
- The empirical results support the efficacy of the proposed method on commonly used VLM benchmarks.

**Weaknesses:**

- One major shortcoming of all presented results is that accuracy and cost are not discussed together. The proposed method reduces the number of visual tokens but adds extra compute and parameters. Therefore, to fairly compare it with other methods, results should show that for the same cost (for example, TTFT), it achieves better accuracy. For instance, in Table 5a it is shown that adding the proposed VR module on top of Qwen2.5-VL improves performance by 1.7. What happens to TTFT and TPS in this setup? Similar information is needed for Tables 3, 2, and 1.
- Missing prior work: In Sections 1 and 2, it is mentioned that existing approaches to tackle the efficiency problem in VLMs involve token pruning or merging, either at the projection layer or in the decoder. However, the authors miss a third category: using vision encoders designed for VLMs that produce fewer visual tokens. For example, the FastViTHD architecture in [1] produces 16× fewer visual tokens than a regular ViT while maintaining accuracy. [1] also discusses multi-level feature utilization. Including a discussion and comparison with this work would clarify the empirical contribution of the proposed method relative to existing approaches.
- In the ablation studies in Section 4: The explanation of why global attention performs worse than the proposed local attention is not very clear. From an efficiency standpoint, local attention is a better choice. However, global attention still has access to local area information, i.e., what can be attended to in global attention is a superset of local attention. Why does it perform worse?
- In Section 3.3 on self-attention: This goal seems orthogonal to the main motivation of efficiency. For example, a simple baseline would be to allow full attention mask for visual tokens. Have you considered comparison with such a baseline?
- The second paragraph in Section 3.3 argues that visual tokens cannot attend to query text tokens. This argument depends on the setup where visual tokens precede text tokens, but this need not be the case. The order could simply be reversed (i.e., text query tokens first, then image tokens). This is common when using interleaved image-text data and does not require an intrusive change to the decoder. Have you considered a baseline where visual tokens come after query text tokens?
- Augmenting visual tokens with the max pooling of text tokens (the text guided token) could potentially break causality: a text token at position k can know some information about text token at position k+1 through attending augmented visual tokens (the output of Vision Remember module). Can you clarify this point?

Minor points:
- Typo in the sentence on lines 242–243.
- Lines 225–226: This is not necessarily an “expansion,” depending on the LLM and vision encoder dimension sizes.

[1] Vasu, Pavan Kumar Anasosalu, et al. “FastVLM: Efficient Vision Encoding for Vision Language Models.” Proceedings of the Computer Vision and Pattern Recognition Conference. 2025.

**Questions:**

- In phase 2, is the image encoder trainable, or is it kept frozen?
- What is the LLM size for the results in this table?
- Section 3.3 on text-guided tokens: What are these text tokens? Are they the original query text tokens, or during inference, when new text tokens are generated (as part of the model’s response), are those also included? From line 470, it appears to be the former.
- Section 1 (the preliminary analysis): Based on the observations, the authors identify text alignment as one of the reasons original visual feature information is lost. But why is this a problem? Forgetting visual features that are not relevant to the query at hand could, in fact, be beneficial.
- In line 360, it is mentioned that for a fair comparison of results in Table 3, the same compression ratio is used. Can the authors provide additional details? The proposed method requires additional parameters as well as extra compute. When comparing with other methods, what compute budget is matched to them?
- When considering global attention in section 4, do you augment visual features with any positional encoding?
- In Figure 2, three dimensions are shown: $D_v$, $D_e$, and $D$. But they are not clearly defined.

---

> ### Author Response · Authors · 2025-11-13
> **Discussion with reviewer TWmZ**
>
> Dear Reviewer, I know from your comments that you have read this article carefully and provided professional feedback. However, some misunderstandings and confusions remain, and I would like to resolve them through discussion, including some details of the article.
>
> 1. Q1:
>
> We keep vision encoder frozen in phase 2. The purpose of our experiment is to verify the effectiveness of the Vision Remember. As long as the settings are the same as the baseline, the only difference is whether or not Vision Remember is added, then a fair comparison can be guaranteed. We find include vision encoder in training could get higher accuracy, but will slow down the experiments, so keep it froze is enough.
>
> 2. Q2:
>
> In Table 1, 2, 4, and 5 (b), we are use Qwen2-0.5B as LLM for fast validation. But our method could achieve accuracy gain on other scale LLM according to Table 2.
>
> 3. Q3:
>
> Yes, text-guided tokens is the hidden states of the inputed user prompt, not the response. Because the LVLM may deal with the visual information in pre-filling stage, not the decoding stage, so the Vision Remember only applies to the pre-filling stage and will not influence the decoding efficiency.
>
> 4. Q4:
>
> The purpose of preliminary analysis is to find where or what could cause the visual information loss, and we identify two reasons. **But the visual information loss is not equal to the accuracy degradation**. Your last sentence refers to the model's accuracy, while our preliminary analysis refers to the loss of visual information; they are not equivalent.
>
> To some extent, I agree with your last sentence “forgetting irrelevant visual information might be beneficial to LVLM accuracy”. But not all forgetting is beneficial. There are many paper find that vision tokens in shallow layers are import, which means the LVLM may deal with the visual information in shallow layers, and **forgetting in these position could harm the LVLM**.  But LVLM pay less attention on vision tokens in deeper layer, and forgetting in deeper layer may be useful.
>
> We also find this phenomenon in the exp. about where we insert the Vision Remember. We insert it after layer 1 and 4 could achieve best performance. If we continue to insert in subsequent layers (compensating for lost visual information at deeper levels), performance will start to degrade.
>
> 5. Q5:
>
> 'The same compression ratio is used' indicates we keep **the retained numbers of vision tokens are the same**, or at least comparable (due to PDrop use hierarchical pruning strategy). For example, we set the compression ratio 1/9 in Tab. 3 (i.e. 64 tokens per patch), then we also retain the 1/9 vision tokens in FastV and VisPruner.
>
> The efficiency comparison with previous methods could be found in the Table 5 (b), and our method achieves the best trade-off on accuracy and efficiency.
> As for latency of 'extra compute', if we remove the Vision Remember, the** TTFT is 100.0ms** and **GPU memory cost is 5140MB**; if we add it, the **TTFT is 104.2ms** and **GPU memory cost is 5320MB**. The results means additional computational delay is only **4.2ms, accounting for 4.0% of the total pre-filling latency**, and the additional GPU memory cost is **180MB (3.4%)**, which is almost imperceptible in actual use.
>
> As for training cost, the training-free methods need a pre-trained LVLM (i.e. LLaVA-NeXT) and then prune the vision token. The training-base methods directly compress the vision token when training the LVLM. From this perspective, our method do not use extra data or scheduler, and the comparisons are still fair. Training the LLaVA-NeXT (pre-trained LVLM of training-free methods) takes 30 hours, while training the Vision Remember takes 6-7 hours, and the acceleration is from less vision tokens.
>
> 6. Q6 and W3:
>
> We find that adding a PE do not significantly change the accuracy for global attention. Even if we don't explicitly add PE in global attention, the visual features come from the Vision Encoder, which means it already contains PE from ViT (implicitly).
>
> As for why global attention could not achieve better performance, we think it because attending to all vision feature may **dilutes** attention, and could not fully leverage the inductive bias and harm the structure information.
>
> 7. W2:
>
> We will add FastVLM in discussion because it has been accepted. Thanks for suggestion.
>
> 8. W1:
>
> The computational latency and memory cost could be found in previous discussion (5), and we will consider your suggestion in the revision.
>
> 9. W4:
>
> I think this experiment is useful but we found it is hard to implement in engineering. Full attention mask need to modify the attention mask and could not be compatible with Flash Attention. A LVLM which could not uses Flash Attention or SDPA is not meaningful and far away from the real production.
>
> 10. W5:
>
> We do not consider this situation because we find most open-source LVLMs (QwenVL, InternVL, LLaVA, MiniCPM) append query text tokens after vision token, and we follow this principle.

---

> ### Author Response · Authors · 2025-11-13
> **Continue the discussion**
>
> Previous page is not enough, but there is still a problem we do not reply, so we add a new page to continue the discussion.
>
> 11. W6:
>
> "The text guided token could potentially break causality" is just speculation, and there is currently no evidence to prove it.
>
> I think it may not a huge problem. The text guided token is only a single token that could not fully contain all text information, and only used in Token-Bidirectional Self-Attention. Then in Token-Feature Cross-Attention, the vision tokens (contain the text guidance) are interaction with vision features, and the interaction mainly work in visual modality, which means that contained text information is further transformed, and the text information in final output may not be understood by subsequent text token.
>
> And the ablation result shows minor accuracy improvement, which to some extent suggests that information leakage is not a major problem, or that there may not be any information leakage at all.
>
> 12. Minor and Q7:
>
> Thank you for reading this article carefully and for pointing out some typos or inconsistencies. We will fix them in the reversion.

---

### Official Review · Reviewer_Vwgw · 2025-10-26

**Soundness:** 3
**Presentation:** 3
**Contribution:** 2
**Rating:** 4
**Confidence:** 4

**Summary:**

The paper studies efficiency in Large Vision-Language Models. Current efficient LVLMs compress or prune vision tokens to reduce cost, but this destroys fine-grained visual cues (text regions, small objects, local layout) and also causes the model to gradually "forget" visual details as text dominates later layers. The paper proposes Vision Remember: instead of only compressing vision tokens once at the projector and hoping the model keeps that info, it repeatedly re-injects (resamples) the original high-resolution visual features back into the LLM decoder layers. It does this with (1) a Token-Feature Cross-Attention layer that performs local cross-attention between current vision tokens and multi-level vision features from early/mid/deep vision encoder layers, and (2) a Token Bidirectional Self-Attention layer that lets vision tokens attend bidirectionally to each other and to a pooled "text-guided token," bypassing causal masks and letting vision attend to the user’s textual reference. The method is inserted between decoder layers, is lightweight, and is meant to be plug-compatible with different projectors and backbones.

**Strengths:**

1. Identified two concrete failure modes in efficient LVLMs (i) information bottleneck from projector compression and (ii) visual cue forgetting across decoder layers. This is a useful diagnosis.
2. Resampling vision features mid-decoder is architecturally simple, does not require retraining the whole LVLM from scratch, and can be attached to different visual projectors and different backbones.
3. Ablations are thorough.

**Weaknesses:**

1. Training cost is not fully discussed. They retrain with CC-558K + 779K instruction tuning. It’s unclear whether Vision Remember needs full two-phase tuning each time you attach it to a new backbone or projector, or whether it can be added with light finetuning on a smaller set. This matters for “plug-and-play” claims.
2. Some methods are re-trained on additional data instead of original released recipes. This might not be fair to training free methods.
3. In the empirical study, the proposed method should also be compared against baselines under multiple compression ratios to further validate its effectiveness.

**Questions:**

1. Does the method require full two-phase retraining every time it is applied to a new backbone or projector?
2. Is it fair to compare Vision Remember with other methods that are retrained on additional data, especially when some baselines are designed to be training-free?
3. Has the proposed method been systematically evaluated against baselines under multiple compression ratios to fully validate its effectiveness across different efficiency settings?

---

> ### Author Response · Authors · 2025-11-12
> **Discussion**
>
> We want to solve some **misunderstandings** here.
>
> 1. Q1, Q2, W1 and W2:
>
> Actually, Vision Remember is **not** a plug-and-play method. It need be trained when attached to a new backbone or projector, and we **did not claim** about "plug-and-play". Our goal is to build a new useful architecture when we train an Efficient LVLM.
>
>  For example, we select LLaVA-NeXT as our baseline, so we **follow the LLaVA-NeXT training data and recipes, rather than loading the pre-trained LLaVA-NeXT weights and training the model with additional data**.
> For other training based methods, we also follow the same recipes and settings, so the comparison is fair.
> For training-free methods, we **first train a LLaVA-NeXT model (without compression) under the same settings, and then add the different pruning strategies based on it**. So the comparison with training-free methods is also fair, because we do not use addition data or longer scheduler.
> In other words, even if some methods claim "training-free", they still need a pre-trained LVLM and then prune the vision tokens. We keep the same settings with their pre-trained LVLM and **keep the same number of vision tokens**, the comparison is fair.
>
> 2. W3 and Q3:
>
> Actually, we do the experiments with **multiple compression ratios**, and the results are showing in **Table 2** and **Figure 1(a)**.

---

### Official Review · Reviewer_3iZ2 · 2025-10-28

**Soundness:** 2
**Presentation:** 3
**Contribution:** 1
**Rating:** 2
**Confidence:** 4

**Summary:**

This paper investigates the causes of visual information loss in large vision–language models (LVLMs) and proposes Vision Remember, a method that re-injects original visual features into decoder layers to mitigate information bottlenecks and visual forgetting.

**Strengths:**

The writing is clear, and the overall presentation is well-structured.

**Weaknesses:**

1. The proposed Vision Remember framework appears to be an incremental improvement over existing LVLM architectures, as it primarily involves a relatively straightforward modification of the self-attention and cross-attention mechanisms. The methodological novelty seems limited without deeper architectural innovation.

2. It is unclear whether Table 1 compares Vision Remember and the baseline under identical training conditions. If both are trained on the same dataset, the practical significance of the improvement is questionable. Given that current multimodal large models, trained on massive image–text corpora, already achieve strong performance on fine-grained tasks such as OCR and Chart & Table Understanding, it remains uncertain whether the proposed method can provide meaningful gains at scale. The authors should clarify the training setup and justify the necessity of this approach under large-scale pretraining.

3. To demonstrate the generality and robustness of Vision Remember, it would be important to evaluate the approach across different MLLMs and model scales, such as Qwen2.5-VL-7B, InternVL.

4. The proposed modules introduce additional cross-attention and self-attention layers, which likely increase both computational and memory costs. However, the paper does not provide any quantitative analysis of this overhead. A detailed comparison of model complexity (e.g., FLOPs, inference time, or parameter count) would help justify the practical trade-offs between performance gains and computational cost.

**Questions:**

See the weakness.

---

> ### Author Response · Authors · 2025-11-12
> **Discussion**
>
> We want to discuss about some misunderstandings here.
> 1. W1:
>
> This work **do not aim to propose a fancy or totally different neural network architecture**. We just want to solve the information loss in Efficient LVLM, so we select the "applications to computer vision, audio, language, and other modalities" track, not other tracks like "deep learning architecture" or "neural network architecture".
>
> **Our novelty lies in (1) we identical two concrete failure modes in efficient LVLMs: information bottleneck and visual cue forgetting (there was no way to study this problem before) (2) we design the attention variant to solve these problem, and the comprehensive experiments demonstrate the effectiveness of our proposed method**.
>
> 2. W2:
>
> We ensure that the **training settings were identical**: the same vision encoder, the same LLM, the same compression ratio, and the same training data and recipe. The **only difference** was whether or not our proposed vision memory was included. Such a completely fair and controllable experiment further proves the effectiveness of our method.
> We will release the code and weights upon acceptance, and if any researchers question the validity of the results, they can reproduce our experiment.
>
> We choose the LLaVA-NeXT as main baseline because it is a **truly and fully open released codebase** (including the training and evaluating code, training data, training recipes and details, and trained weight). To date, **many accepted works have chosen LLava-NeXT as their baseline**, making it one of the most widely used codebase in academia. Validating our method on **the most commonly used academic baselines** is sufficient. If a company endorses our method, they can use Vision Remember to improve their LVLM and validate it on large-scale, proprietary, massive image-text corpora. Not all academic researchers have the **vast computational resources** and **unlimited data** to support such requirements.
>
> 3. W3:
>
> We report the experimental results on **two different baseline: Qwen2.5-VL and MiniCPM-V** in Table 5 (a), and the **accuracy gains** have demonstrated the generalizability and robustness of Vision Remember. There are so many LVLMs in the community, and it is time-consuming and unreasonable to require us to validate all methods.
>
> 4. W4:
>
> Although the additional modules introduced increase computational overhead, our local attention approach allows for efficient cross-attention computation, as discussed in Section 3.2. Furthermore, our designed modules are **compatible with acceleration operators such as SDPA**, further improving computational efficiency. Furthermore, we **only insert the vision remember after two layers**, which increases the computational burden less compared to dense insertion.
>
> After we tested on a single A100 GPU, **the TTFT of baseline is 100.0ms, and the TTFT of our method is 104.2ms**. The additional computational delay is only **4.2ms**, accounting for **4.0%** of the total prefilling latency, which is **almost imperceptible** in actual use.
> And the **decoding speed did not increase** because Vision Remember only applies to the pre-filling stage.
> The GPU memory cost of baseline is **5140MB** and that of our method is **5320MB**, and adding Vision Remember will only increase GPU memory by **180MB (3.8%)**.

---

### Official Review · Reviewer_KgXE · 2025-10-31

**Soundness:** 3
**Presentation:** 3
**Contribution:** 2
**Rating:** 4
**Confidence:** 4

**Summary:**

This paper proposes Vision Remember, a module to recover visual information lost in efficient Large Vision-Language Models (LVLMs). Traditional LVLMs compress vision tokens for efficiency but lose fine-grained spatial cues. Vision Remember resamples original visual features across LLM decoder layers through two components: a Token-Feature Cross-Attention Layer for local feature resampling and multi-level fusion, and a Token Bidirectional Self-Attention Layer for bidirectional visual-text interaction. Experiments on 11 benchmarks show consistent improvements (up to +5.7 over baselines) with better efficiency, demonstrating its generality across various LVLMs and vision projectors.

**Strengths:**

1. The overall writing is relatively clear.

2. Experiments show consistent improvements with better efficiency, demonstrating the effectiveness of the module.

**Weaknesses:**

1. The phenomenon shown in Fig. 1b is not particularly new, it has been reported repeatedly (e.g., FastV, PyramidDrop).

2. The performance of “LLaVA-NeXT” is too low; validating an algorithm on it is therefore not very convincing, you can build stronger baselines with more data and new LLM.

3. Are the FastV and PDrop configurations presented in Table 3 their default settings? Please also show the results with no compression applied.

**Questions:**

see weakness

---

> ### Author Response · Authors · 2025-11-12
> **Discussion**
>
> We want to solve some **misunderstandings** here.
>
> 1. W1:
>
> Actually, our preliminary analysis is not as the same as in FastV and PDrop.
> In the Figure 1 of FastV, they use the different pruning ratios to demonstrate **the performance of the LVLM degrade with the less retained vision tokens**. In the Figure 3 and Figure 4 of FastV, they demonstrate **the LVLM may not pay more attention on vision tokens in middle and shallow layers, which means these vision tokens may not be important and could be pruned, and due to the vision tokens occupy the majority, pruning them could improve the efficiency**. As for Figure 1 of PDrop, they also shows the LVLM performance with different pruning ratios and pruning positions, and their conclusion is **LVLMs are sensitive toward token dropping on shallow layers, regardless of the dropping ratio**, so they propose a hierarchical pruning strategy and do not drop the vision token in shallow layers.
>
> Different from FastV and PDrop, we use **linear probing** and show the **classification accuracy on Tiny-ImagNet**. Linear probing could probe the visual information, and so our preliminary experiment is mainly used to **evaluate the  where and how extend the visual information is loss**. Our analysis **identify two concrete failure modes in efficient LVLMs (i) information bottleneck from projector compression and (ii) visual cue forgetting across decoder layers.** And the analysis is also approved by other reviewers (**reviewer Vwgw said 'useful diagnosis' and reviewer TWmZ said 'linear probing results is interesting'**)
>
> 2. W2:
>
> We choose the LLaVA-NeXT as main baseline because it is a **truly and fully open released codebase (including the training and evaluating code, training data, training recipes and details, and trained weight)**. To date, many **accepted works** have chosen LLava-NeXT as their baseline, making it **one of the most widely used** codebase in academia. Build upon this codebase, we could easily make comparisons and our work could be reproduced by others after code releasing (when this paper is accepted, we will release the code and weights).
>
> However, many newer LVLMs are trained on **private datasets** using **a lot of computational resources** and **only release the inference code and model weights**, making the complete training process **impossible to reproduce** in academia. Using more data, new LLM models, and new model architectures makes it easier to **question the fairness of comparisons**.
>
> Besides, we also make experiments on **two different baseline**: Qwen2.5-VL and MiniCPM-V, and the results are shown in Table 5 (a).
>
> 3. W3:
>
> When we evaluate the FastV and PDrop, we do not use their default settings (mainly compression ratio). In Table 3, the vision remember is tested on a compression ratio 1/9, so we retain the 1/9 vision tokens in layer 3 on FastV (pruning position is the same). We also make similar changes to PDrop to **ensure that the remaining vision token quantities are comparable**, thus achieving a **fair comparison**.
>
> The result with **no compression** applied (i.e. LLaVA-NeXT with the same vision encoder, LLM, training data and recipes) is shown in Table 5 (b), and the average score is **49.8**. Our method with compression ratio of 1/4 achieves **49.7**, only 0.1 drop. But our approach is more efficient. And our method also **outperform** the VisPruner (representing the training-free pruning) and TokenPacker (representing the training-base compression) with a large margin.

---

### Note · Authors · 2025-11-14

I have read and agree with the venue's withdrawal policy on behalf of myself and my co-authors.